# New Calamitic Mesogens Exhibiting Aggregation-Induced Emission (AIE)

**DOI:** 10.3390/ma17143587

**Published:** 2024-07-20

**Authors:** Saurav Paul, Bimal Bhushan Chakraborty, Nandiraju V. S. Rao, Sudip Choudhury

**Affiliations:** 1Department of Chemistry, Assam University, Silchar 788011, India; chemsaurav1@gmail.com (S.P.); bimalbhushan11@gmail.com (B.B.C.); drnvsrao@gmail.com (N.V.S.R.); 2Centre for Soft Matter, Department of Chemistry, Assam University, Silchar 788011, India

**Keywords:** AIEgen, liquid crystal, Schiff’s base

## Abstract

Aggregation-induced emitters or AIEgens are generally signified by their stronger photoluminescence in aggregation than in the solution state. Due to high emission efficiency in aggregate and solid states and good processability, organic AIEgens drew attention to the development of advanced luminescent materials. However, as mesogenic materials self-assemble to a different molecular arrangement in different phases, achieving liquid crystallinity and AIE properties in the same molecule would provide a valuable tool to develop solvent-independent AIEgenic materials. With this goal, the present work reports the synthesis of new organic thermotropic liquid crystalline compounds exhibiting aggregation-induced emission (AIE). The synthesized compounds exhibit strong green luminescence in a solid state which sharply quenches upon entering smectic mesophase by heating. This is in addition to the exhibition of dispersion medium (solvent)-dependent emission, thus providing a dual mode of AIE. The mesogenic property of the synthesized compounds was studied by XRD, POM, and DSC. The AIE was studied by fluorescence spectroscopy and variable temperature fluorescence microscopy. A DFT study was carried out to gain an insight into the AIEgenic behavior of the material.

## 1. Introduction

Since the discovery of the phenomenon of aggregation-induced emission (AIE) in 2001, luminogens exhibiting aggregation-induced emission (AIEgens) have gained much interest [1]. Such luminogens have emerged as the advanced material of choice with applications in various fields such as chemical sensing [2,3], light emitting devices, bio-imaging [4,5,6,7], etc. AIEgens show very weak emission in solution but emit strongly in an aggregated or solid state. They often exhibit unique advantages like high photostability, high solid quantum yield, and no aggregation-caused quenching (ACQ) effect, thereby offering very promising candidacy in fields such as OLEDs, stimuli-responsive sensing, and theranostics [8,9,10,11]. Various organic cores, such as tetraphenylethene (TPE) [12,13,14], Triphenylamine [15], Tolane [16], cyanostilbene [17], diazafluorenone [18,19], etc., have been employed to design AIEgens. Apart from organic compounds, AIEgens containing transition metals, such as Cu(I) [20], Zn(II) [21], Ir(III) [22], Pt(II) [23], Au(I) [24], etc., have been reported. A comprehensive review of transition metal-containing AIEgenic materials was published by Alam et al. [25]. Recently, Voskuhl and Giese [26] reported a very informative review on mesogens with AIE properties.

The design of compounds co-exhibiting AIE and mesogenicity is quite challenging. This is because the way molecules aggregate is a key factor in both AIE (aggregation-induced emission) and liquid crystallinity [26]. However, designing molecules with the right balance of properties for both mesogenicity and AIE is challenging. While there are reports of mesogenic compounds with good fluorescence [27,28,29,30,31,32], there are very few reports of mesogens that exhibit AIE. 

After Yuan’s report in 2012 for the first time about the AIE behavior of TPE-based mesogens [33], the majority of the AIE-based mesogens have been designed around the tetraphenylethene (TPE), cyanostilbene (CST), and tolane (TOL) as the emitting motif [26]. AIE-based liquid crystalline materials have been synthesized using 3-[4-(diphenylamino)phenyl]acrylonitrile [34], pentylcyclohexyl [35], and methacrylate [36]. Aggregation-dependent emissions of H-bonded mesogen have recently been reported by our group [37]. Usually, to design mesogenic materials that exhibit AIE behavior, three approaches are adopted, viz., (i) the covalent attachment of AIEgenic emitter to mesogenic compounds, (ii) the doping of a mesogen with an AIEgen, and (iii) the formation of suitable supramolecular assemblies employing noncovalent interactions, (like metal/ligand interactions, hydrogen bonding, charge transfer, etc.). Amongst these three, the supramolecular approach is, so far, the least investigated one [26]. 

With continued interest in the design of AIE exhibiting mesogenic compounds based on structurally simpler calamitic architecture, here we report a new fluoro-terminated single side chain Schiff base mesogen exhibiting AIE property. 

## 2. Materials and Methods

### 2.1. Materials and Instruments

All the chemicals were purchased from Sigma Aldrich (St. Louis, MO, USA) and used as obtained. Solvents were procured from Merck (Rahway, NJ, USA). Polarized optical microscopy (POM) was performed on a Nikon LV100 equipped with Instec Hotstage (Nikon Instruments Inc., Melville, NY, USA). DSC, FTIR, UV-Visible, powder XRD, and Elemental analysis were conducted on PerkinElmer Pyris 1 (PerkinElmer, Waltham, MA, USA), Bruker Alpha II (Bruker, Billerica, MA, USA), Jasco V730 (Jasco, Oklahoma City, Oklahoma), Rigaku Ultima IV (Rigaku, Tokyo, Japan), and PerkinElmer 2400 CHN Analyser, respectively. 

### 2.2. Synthesis of the AIEgenic Compounds

The synthesis procedure for the target AIEgenic compounds is outlined in Figure 1. Alkoxy benzoic acid (1) was prepared by the alkylation of Ethyl 4-hydroxybenzoate by Williamson ether synthesis followed by hydrolysis. Alkoxy benzoic acid so obtained was esterified with 2,4-Dihydroxybenzaldehyde using DCC/DMAP to obtain 4-formyl-3-hydroxyphenyl, 4-alkoxybenzoate (2). This compound (2) was then treated with 4-fluoro-3-nitroaniline to obtain the target compound (4F3NAn, n = 12, 14, 16). Synthetic details and spectroscopic analysis are presented in Appendix A.

## 3. Results

### 3.1. Absorption Spectroscopy of the Synthesised Compounds 

The absorption spectrum of the compounds is shown in Figure 1. The UV-Vis spectrum of 4F3NA12 has been recorded in different solvents like hexane, ethylacetate, ethanol, chloroform, and acetonitrile and we found two absorption bands at 270 nm and 330 nm, respectively, in every case (Figure 1a). Absorption spectra of all the compounds (as chloroform solution) are shown in Figure 1b.

### 3.2. Fluorescence Study of the AIEgens

The solid-state fluorescence spectra of the compounds were recorded with the Ocean Optics QE65Pro spectrometer coupled with a - sample stage through optic fiber. All the compounds of the 4F3NAn series exhibited strong fluorescence at around 550 nm in solid state upon excitation with 330 nm radiation. The fluorescence disappeared completely when the compounds were dissolved in chloroform. The fluorescence spectra of the compound 4F3NA12 in the solid state are shown in Figure 2a. The solid-state fluorescence spectra of the other compounds are presented in Appendix A. The image of the solid-state emission under a UV chamber (Ex 365 nm, broadband) is shown in Figure 2b. The strong fluorescence of the compound (in the solid state) was quenched upon dissolution in chloroform. Then, ethanol was used as a solvent where the compound was sparingly soluble. It exhibited some weak fluorescence arising from the tiny solid particles suspended in the solvent. Figure 2c represents the visual image of the compound in a UV chamber showing no emission from the clear chloroform solution, and some fluorescence in EtOH where the compound is sparingly soluble. This indicates that the compound loses its luminescence property in fully solvated conditions and the aggregation of the molecules is necessary for fluorescence to occur. 

The AIE effect in the compound 4F3NA12 was studied using two solvents, chloroform and ethanol, of varying ratios. As Schiff’s base is incompatible with water, hence water was not used as a solvent. Since the compounds dissolve in chloroform but not ethanol, the solutions of 4F3NA12 were prepared in chloroform/ethanol mixtures with different volume/volume (*v*/*v*) ratios of ethanol, ranging from 0% to 100%, and the fluorescence spectrum of each solution was recorded with an excitation wavelength of 330 nm (Figure 3a). An increase in the ethanol fraction of >70% in the solvent mixture triggered a sharp increase in the fluorescence intensity indicating the onset of aggregation-induced emission. A higher ethanol fraction caused a significant increase in fluorescence, thereby confirming the AIE property of the compound (Figure 3b).

### 3.3. FT-IR Study of 4F3NA12

The FT-IR spectrum of 4F3NA12 as a representative compound is shown in Figure 4. The two peaks nearly at 2915 cm^−1^ and 2846 cm^−1^ appear which are attributed to the alkyl chain of 4F3NA12 indicating the existence of the long alkyl chain in the synthesized organic molecule. These two peaks arise due to the asymmetric and symmetric –CH stretching mode of CH_2_ groups, respectively. The characteristic C=O stretching of the ester group appeared at 1735 cm^−1^.The peak at 1600 cm^−1^ affirms the presence of the imine group in the system.

### 3.4. Polarized Optical Microscopy 

The optical textures of 4F3NAn (n = 12, 14, 16) were investigated using polarized optical microscopy connected with a hot stage to explore the characteristic textures and mesomorphic phases (Figure 5a–f). All the synthesized compounds in this series were found to be stable during the course of repeated heating and cooling. 

When the compound 4F3NA12 was heated to 100 °C, the material melted and formed a typical thread-like smectic A texture (Figure 5b). This mesophase remained stable until 173 °C, at which point it transformed to an isotropic state. In the cooling cycle, as it cooled down from isotropic melt, the focal conic fan texture (Figure 5c) co-existed with regions of homeotropic alignment (molecules perpendicular to the plane of observation), and both characteristics of the SmA phase appeared. The texture persisted until it crystallized at 74 °C. In the mesophase, the presence of the focal conic fan texture suggests a layered arrangement, and the homeotropic region indicates molecular orthogonality with respect to the layer planes [38], which are characteristics of the smectic A phase. 

The other two compounds of the series, 4F3NA14 and 4F3NA16, also showed similar liquid crystalline behavior. The SmA to crystal transition (in 4F3NA14), isotropic to SmA transition through *bâtonnet* growth (in 4F3NA16), and co-existing focal conic and homeotropic regions of SmA (4F3NA16) are presented in Figure 5d, Figure 5e, Figure 5f, respectively.

### 3.5. Differential Scanning Calorimetry (DSC) of the Mesogens

All the synthesized mesogens exhibited two transitions during both the heating and cooling cycles in DSC run at 5 °C/min under N_2_ atmosphere and the observations are consistent with the results obtained from the POM studies. The DSC thermogram of the representative compound 4F3NA12 is presented in Figure 6. The thermograms of all the compounds can be found in the Appendix A. During the heating cycle, two endothermic peaks at 97 °C and at 176 °C correspond to the Cr-> SmA and SmA-> Iso transitions, respectively. Upon cooling, an exotherm with an onset temperature of 174 °C marks the transition of SmA from the isotropic melt, followed by another peak at 70 °C corresponding to the crystallization from the mesophase. The enthalpy change associated with the Cr-SmA transition was found to be significantly higher compared to that of the SmA-Iso transition. This observation suggests a substantially higher degree of disorder between the crystalline phase and the SmA mesophase compared to the disorder between the SmA and isotropic phases. Phase transition temperatures as per the DSC analysis for all the compounds in the series are presented in Table 1.

### 3.6. Variable Temperature X-ray Diffraction 

X-ray diffraction measurements were carried out on the compound 4F3NA12 to explore the mesophases and study the geometric structure parameters. The powder XRD patterns of an unaligned sample 4F3NA12 during cooling from isotropic melt exhibited sharp signals in the small-angle region (Figure 7a) and a lower intensity broad peak in the wide-angle region (Figure 7a Inset). Figure 7b represents the change in the XRD pattern of the virgin sample at different temperatures. The sharp intense peak (Figure 7a) corresponds to layer ordering with layer spacing, d = 35.6 Å, which is larger than the calculated molecular length, L = 31.4 Å (single molecule, gaseous state, DFT optimized). The experimental layer thickness is found to be 1.13 times the theoretical molecular length, indicating a partial bilayer nature in the SmA phase (Figure 8a). Different plausible partially overlapped SmA arrangements are presented in Figure 8b–e. However, keeping the facts that the d-L of about 4.2 Å (also, there is a possibility of short-range intermolecular H-bonding between O-H as HBD and NO_2_ as HBA), the arrangements 7d and 7e were ruled out. Similarly, the continuous array of H-bonded ensemble as in Figure 8c seems improbable because such extensive lateral hydrogen bonding would significantly tilt the molecules, leading to L > d, which contradicts the XRD results and POM data. The H-bonded dimeric molecular arrangement represented in Figure 8a and schematically shown in Figure 8b aligns well with the XRD data. The broad diffuse reflection in the wide-angle region corresponding to 3.91 Å indicates the expected fluidic nature of the LC phase. In Figure 7b, the room temperature XRD pattern was recorded with the virgin powder sample with randomly oriented crystals. Then, the temperature was raised to 120 °C, equilibrated for 5 min, and the XRD pattern was recorded to detect the SmA mesophase. The temperature was then gradually decreased to 100 °C, 80 °C, and 40 °C, and the XRD pattern was recorded after 5 min equilibration at each case. During the process of crystallization from a semisolid mesophase (inside the sample holder), the preferred orientation of the tiny newly grown crystals might have taken place. That is why the XRD patterns recorded with the virgin sample at room temperature and that taken during the cooling cycle at 40 °C exhibited different peak ratios.

### 3.7. Variable Temperature Fluorescent Microscopy of the AIEgens

The aggregation-induced emission (AIE) properties of the synthesized organic mesogens were studied using a temperature-controlled fluorescent microscope. The microscope was interfaced with a spectrofluorometer via an optic fiber cable. During the absorption study, all the compounds (4F3NAn) in this series exhibited their first absorption around 330nm. Under the fluorescent microscope, the virgin sample at room temperature (solid state) exhibited green fluorescence (emission λmax around 550 nm) upon irradiation through a 300–360 nm band cube filter. As the temperature increased, the fluorescence of the sample was sharply quenched upon the onset of the SmA phase. Further increase in the temperature resulted in the transition of the sample to an isotropic state, remaining non-fluorescent. All three compounds exhibited the same emission behavior under the fluorescent microscope. Figure 9 demonstrates the crystalline texture (at room temperature) and SmA texture (at 155 °C) of 4F3NA12 under crossed polarizers and the corresponding fluorescence microscopy image maintaining the sample position unchanged. 

The observed green emission in the solid state originates primarily from the core structure of the material where a number of aromatic cores have been connected through conjugation. The structure contains an electron donor and acceptor group which provides the system the ability to show fluorescence in the solid state. It has been observed that the presence of donor/acceptor groups in two ends or in the middle of such organic core offers greater control over their fluorescence property [39]. In the solid state, the collaterality of the two imine-connected terminal rings (necessary for the exhibition of the fluorescence) is maintained due to the restriction of intramolecular rotations (RIR). However, in the mesophase, the molecule enjoys more vibrational and rotational freedom. Different parts, specifically the aromatic rings of the molecule, could rotate about a single bond. Such rotations remove the restriction of intramolecular rotations (RIR). The rotation of the terminal aromatic rotor dissipates the excited-state energy, letting the absorbed energy decay without luminescence, thereby quenching the fluorescence. Further, π-π interaction in the solid state of this molecule may also augment the emission process by decreasing the HOMO-LUMO energy barrier, as reported in other systems [40].

### 3.8. Computational Study

To gain an insight into the AIEgenic behavior exhibited by the 4F3NAn series mesogens, DFT, TD-DFT, and molecular dynamic studies were performed on the smallest member of the series, 4F3NA12. The structure was optimized by the GAUSSIAN 09 software [41] under DFT at the level of B3LYP with basis set 6-31G(d) [42,43] and visualized using the Avogadro [44,45] suite. The optimized structure (Figure 10a) showed the imine-connected aromatic rings A and B to be in the same plane providing efficient pi conjugation. Ring C is not co-planner with A and B due to the ester linkage and hence, does not effectively contribute to the pi conjugation of the other two rings. Therefore, it is assumed that the co-planner, conjugated, and intramolecular H-bonded rigid part of the molecule (containing rings A and B) is responsible for the exhibition of fluorescence, and therefore, any shift of this coplanarity would lead to the quenching of the luminescence. 

To model the effect of elevated temperature on the conformation, the molecule was subjected to molecular dynamics under an MM2 force field with a 2.0 fs step interval. After equilibration at 300 K (25 °C), the temperature was allowed to rise up to 373K (100 °C) at a heating rate of 1.000 Kcal/atom/ps. The simulation (of 5000 steps) was performed with a single molecule in the gas phase to eliminate all the possible restrictive interactions (intermolecular interactions, solvent effects, etc.) and to allow maximal conformational changes. The interplanar angle between the rings A and B was observed to vary between 0° and 63°, and no instance of the coplanarity of ring C with ring B was detected within the simulation timeframe.

Keeping the simulated range of the interplanar angle between the rings A and B in view, six structures were prepared from the original optimized structure with different A/B interplanar angles. The interplanar angle between rings A and B was changed to 15°, 30°, 45°, 60°, 75°, and 90° keeping the other structural features unchanged to prepare the structures D15, D30, D45, D60, D75, and D90 (Figure 10b). The absorption spectrum of all seven structures was calculated through DFT with the TD-SCF method. The energy profile of the important orbitals is presented in Figure 10c. 

The computational analysis of the optimized structure correctly predicted two absorption peaks similar to the experimental observation. The difference between the peak positions between the computed and experimental spectrum seems to be due to the different physical conditions involved and the donor/acceptor-type architecture of the molecule. However, as the objective here was only to observe the trend in the absorption with the change in the A/B interplanar angle (Appendix A), this difference was not significant. The molecular orbitals (MOs) of the 4F3NA12 molecule (optimized structure) are illustrated in Figure 10d. The electron contour shows that the MO 142 (HOMO) is mostly contributed by the conjugated pi system of the rings A and B. The MO 143 (LUMO) is formed from exclusive contribution from the atomic orbitals from ring A. The first excitation (460 nm) is solely due to the HOMO → LUMO transition with a coefficient of 0.69708. The second excitation (393 nm) is due to three transitions 138 → 143 (Coeff 0.12717), 140 → 143 (Coeff 0.18488), and 141 → 143 (Coeff 0.66654), with the 141 → 143, i.e., (HOMO-1) → LUMO transition being the most dominant one. The excitation from 140 → 143 and 141 → 143 involves intramolecular charge transfer to LUMO from HOMO-1 and HOMO-2, respectively. With the gradual increase in the A/B interplanar angle, the first excitation involving 142 → 143 (HOMO → LUMO) transition exhibited a gradual blue shift due to a reduction in coplanarity between the rings A and B, resulting in a less conjugated pi system. At the same time, the oscillator strength steadily increased for the second transition primarily due to the 140 → 143 and 141 → 143 charge transfer transitions (Table 2 and Figure 10c,d).

## 4. Discussion

All the compounds in this series exhibited strong fluorescence in the solid state, and they were non-fluorescent when dissolved completely in a solvent (like chloroform). In ethanol, where the compound is only sparingly soluble, it exhibited low fluorescence arising from the tiny solid particles suspended in the solvent medium. The observation affirms the AIEgenic behavior of the compound.

All three compounds reported in this work exhibited thermotropic liquid crystalline properties. Upon heating, the compounds in the series transitioned from the crystalline solid phase to the smectic A mesophase (confirmed by XRD), which transformed to the isotropic state upon further heating. The phase sequence shown by the compounds was reversible. When the fluorescence properties of the compounds at different phases were investigated with variable temperature fluorescence microscopy, it was revealed that the fluorescence observed at the crystal (solid) state sharply quenches upon transition to the liquid crystalline state. It indicates that the difference in the molecular arrangement between the crystalline and SmA mesophase was sufficient to trigger the onset of the emission or the quenching of it (Figure 11a). 

The molecular arrangement in the mesophase was studied with variable temperature XRD. The XRD analysis affirmed the partial bilayer nature in the SmA phase of the compound. The smectic layer was found to be 1.13 times the theoretical molecular length. This indicates the existence of a substantial pi-pi interaction in the mesophase to maintain the partial bilayered smectic self-assembly. Such a strong pi-pi stacking interaction could be detrimental to the AIE phenomenon [46] and might be responsible for the observed quenching of fluorescence in the liquid crystalline state.

The computational study showed that with the gradual coplanarity between ring A and ring B, the first excitation involving the 142 → 143 (HOMO → LUMO) transition exhibited a gradual blue shift due to a less conjugated pi system. At the same time, the oscillator strength steadily increased for the second transition primarily due to the 140 → 143 and 141 → 143 charge transfer transitions (Table 2 and Figure 10c,d). As reported by Hudson, such charge transfer transitions can reduce the fluorescence efficiency of the system [47].

Keeping that in view, dimeric systems with antiparallel, parallel with -NO_2_ groups in the same side, and parallel with -NO_2_ groups on the opposite side (coded as AntiParallel, P-SynNO2, and P-AntiNO2, respectively) were prepared from the original optimized structure (Figure 10b). A gap of 3.6Å was kept between the planes of ring B of the two molecules for effective intermolecular pi-pi interaction [48] and the absorption profile was computed. All three systems exhibited sharply lowered HOMO-LUMO gap (Figure 10c) and consequently, a significant red shift (about 22% decrease in absorption wavelength) in absorption maxima (Table 2) compared to those in the single molecule, D0 (the electron map of all the HOMO and LUMO’s are consolidated in Appendix A with the DOS profile of the D0 in Appendix A). 

As mentioned earlier, in the solid state, the required degree of collaterality of the rings A and B necessary for the exhibition of the fluorescence is maintained due to the restriction of intramolecular rotations (RIR). In contrast, a fully solvated environment grants rotational freedom to the terminal aromatic ring. The rotation of the terminal aromatic rotor dissipates the excited-state energy, letting the absorbed energy decay without luminescence—similar to a typical AIEgen, like tetraphenylethene (TPE). However, for the compounds under study, the liquid crystal state might not permit the complete free rotation of the terminal ring due to the inherent self-assembly of the molecules. Here, in the mesophase, the rotational twisting of the terminal aromatic ring (Figure 11b) seems to be enough to quench the emission due to (i) a significant reduction in the pi-system conjugation as evidenced by a marked increase in the HOMO/LUMO gap, (ii) pi-pi stacking in the mesophase, and (iii) the increased dominance of the charge transfer transitions reducing the fluorescence efficiency [47]. Additionally, the in-plane/out-of-plane group-swinging vibrational motions may have augmented the non-radiative decay of the excited-state energy [49].

## 5. Conclusions

A new series of organic AIEgenic liquid crystalline compounds based on a simple calamitic framework have been successfully synthesized and reported. All the members of the series showed aggregation-induced emission. The compounds emitted green fluorescence in the solid state and their emission disappeared upon solvation or upon entering into the mesophase by the elevation of temperature. In the synthesized compounds, the emission seems to be very sensitive to molecular aggregation; indeed, the molecular self-assembly of the smectic liquid crystalline arrangement was insufficient to retain the fluorescence observed in the solid state. Due to the intricate interplay of noncovalent interactions governing aggregation-induced emission (AIE) and the dependence of AIE properties on a molecule’s rotational freedom and intramolecular vibrations, merging these fascinating material characteristics—AIE and liquid crystallinity—within a single molecule holds immense promise for the development of advanced materials with diverse applications.

## Data Availability

The data presented in this study are available in the Appendix A.

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
