# Peer review of "New Calamitic Mesogens Exhibiting Aggregation-Induced Emission (AIE)"

_materials, 2024, doi:10.3390/ma17143587_

Round 1

Reviewer 1 Report

The subject of this work is interesting but there are several experimental weak points that must be addressed by the authors. There are also some English words and expressions to check.

Line 25 the word coining is quite odd for me

Line 27/28 various is repeated

Line 33 PTE tetraphenilethene or ethylene?

Line 47 small is not the appropriate adjective

R Line 108 I do not understand representee

Figure 1 UV spectrum in Hexane shows negative values, I think it is necessary to repeat the measurement; the same for the emission in chloroform reported in Fig 2.

Figure 4 the quality of the images is poor, hence is difficult to discern the texture in Fig. 4b. Nevertheless it seems an oily streak texture (sometimes reported for Smectic A see Dierking) instead of a thread-like texture (could you provide a reference?). In the caption and in table 1 (4F3NAC12-4F3NAC14, 4F3NAC16) the sample is named 4F3NAC12 instead of 4F3NA12. Please uniform the names also in the supporting information. The description in lines 127-128 is not very clear, please clarify using the reference 38 cited.

DSC measurements: in the supporting the measurements of 14 and 16 are reported. The captions are not correct, please check. Figure S8 should be 4F3NA16.

In FigureS9 (4F3NA14) there are three peaks. Could you explain the peak at 102.72°C for this sample?

By the way, No transition to Nematic phase is observed, is it right?

In table 1 the errors in the measurement and the correct approximation should be reported (i.e. 177.76 °C according to supporting should be 177.77°C). The names of compounds contains C (4F3NAC12).

X ray measurements: the measurements of 4F3NA14 and 4F3NA16 are not reported. Furthermore the scale in Figure 6 should start from 2.5 degrees and show more dense abscissa to highlight the peak region. No details are provided for the geometry of the measurements and no second order peak is observed for the smectic A phase. Could you give an explanation? Figure 6b is not commented in the text.

Have you any information about the structure of the Cr phase? Apart the X ray pattern reported in figure 6b?

Row 217 Avogadro not Avogedro

Lines 272/273 please check the sentence

The comments on the quality of English are reported previously. Some expressions must be checked.

Reviewer 2 Report

The work of Saurav Paul et al. is devoted to synthesis and investigation of a new series of mesogenic compounds. The compounds possess the smectic-A liquid crystal phase in a certain temperature interval between crystal and isotropic phases. Standard experimental methods such as polarised optical microscopy, FTIR, optical spectroscopy, DSC, XRD were employed. DFT calculations were performed with the aim to understand the observed optical properties. 

The primary interest in the synthesized compounds is their photoluminescent behavior. The materials show strong fluorescence in the solid state, but no emission in good solvents and in the liquid crystal state. The authors discuss the observed behavior on the basis of experimental data and resuts of calculations.

The obtained results might be interesting for specialists working in the field. I have no objections against publication of essential results, but my opinion is that the manuscript needs a revision. My specific comments are as follows.

1) I cannot fully understand Figure 1. First of all, the caption indicates it shows UV-Vis spectra, but the graph only extends to 400 nm, so it is more like UV only. The curve labeled "Hexane" seems to show absorbance below zero, which is odd and may indicate uncalibrated measurement.  

2) What is the reason of the substantial change in the XRD pattern between room temperature and 40oC in Figure 6b?

3) The text in lines 35-39 in page 1 is confusing and has to be rewritten.

4) Wording in some places of the manuscript is not clear. For example, in lines 331-333 the authors write: "All the members of the series showed aggregation-induced emission, which can be triggered either by heat or by solvent". The reader might think that emission is initiated by heating or by dissolution of the material in some solvent. In fact, as written in the next sentence, the situation is the opposite: emission is strong in the solid state, but disappears on heating and in the solvents.

 In summary, I recommend a revision of the manuscript.

Some sentences in the manuscript are hard to understand. I have an impression that some editing of English is necessary. 

Reviewer 3 Report

Comments on materials-2554900.

In this work, S. Paul et al. have synthesized a series of compounds possessing both AIE characteristics and liquid crystallinity, and their properties were systematically examined. The subject matter of this study holds intrinsic interest and is poised to furnish valuable insights to fellow researchers in this research field. However, before contemplating the publication of this work in the ‘Materials,’ several crucial revisions are imperative:

1. Primarily, it is pivotal to elucidate the rationale behind designing these intricate molecular structures. While a series of compounds have been synthesized with diverse modifications to the alkyl chain within the core unit, the current manuscript predominantly centers around the investigation 4F3NA12. Regrettably, exploring the structure-property correlation governed by the alkyl chain variations remains conspicuously absent.

2. The absorption spectrum for the n-hexane solution in Figure 1 necessitates re-measurement, and it is imperative to incorporate the concentration details of the solution.

3. The absorption spectra for 4F3NA14 and 4F3NA16 were not included in the main text or SI. Their experimental results should be included. In addition, the absorption spectra in the solid state of each compound should be included.

4. Conventional practice for investigating the AIE phenomenon often entails scrutinizing alterations in emission characteristics upon introducing water to a solution of THF or acetone, typically in varying ratios. Consequently, a meticulous exploration of AIE attributes via this experimental paradigm is highly recommended.

5. The results and discussion for the 4F3NA14 and 4F3NA16 on the POM are missing.

6. The authors assert that 4F3NAn compounds constitute a dyad system encompassing electron-donating and electron-accepting moieties within their molecular structure. Consequently, it becomes imperative to integrate electrochemical property experiments into the comprehensive analysis.

Moderate editing of English language required.

Round 2

Reviewer 1 Report

Unfortunately I was not able to download the new supporting information file.

In the caption of Fig.5 there are some 4F3NACn labels left

line 154 please correct homaeotropic

table 1 I asked in the previous report to give some information on the errors which affect the measurements.

X ray diffraction I still do not understand the utility of fig. 7b, since it is not commented in the text. If you want to show the crystallinity of the compound a function of the temperature, you should include the low angle region as well and discuss in the paper the result. By the way, In my previous report I did not ask about a single crystal measurement but a powder one.

no particular comments

Reviewer 2 Report

In the revised version of the manuscript the authors made changes in accordance with comments and questions of the reviewers. Data on new measurements are added. Unclear sentences have been rewritten.

In my opinion, the manuscript can be accepted for publication.  

Author Response

We appreciate you for your precious time in reviewing our paper and providing valuable comments. Thank you.

Reviewer 3 Report

The authors adequately answered the reviewers’ comments and revised the manuscript. This reviewer believes it is now acceptable in Materials. 

Author Response

(The authors gave the same response as above.)
